# The Influence of bFGF on the Fabrication of Microencapsulated Cartilage Cells under Different Shaking Modes

**DOI:** 10.3390/polym11030471

**Published:** 2019-03-12

**Authors:** Xia Zhou, Xiaolin Tang, Ruimin Long, Shibin Wang, Pei Wang, Duanhua Cai, Yuangang Liu

**Affiliations:** 1College of Chemical Engineering, Huaqiao University, Xiamen 361021, China; josiechou1993@163.com (X.Z.); xmtangxiaolin@163.com (X.T.); simplever@126.com (R.L.); sbwang@hqu.edu.cn (S.W.); wang_pei_yes@163.com (P.W.); Leaduanhua@163.com (D.C.); 2Fujian Provincial Key Laboratory of Biochemical Technology, Huaqiao University, Xiamen 361021, China; 3Institutes of Pharmaceutical Engineering, Huaqiao University, Xiamen 361021, China

**Keywords:** 3D micro-gravity, cartilage tissue, bFGF, artificial cells, microcapsules

## Abstract

Cell encapsulation in hydrogels has been extensively used in cytotherapy, regenerative medicine, 3D cell culture, and tissue engineering. Herein, we fabricated microencapsulated cells through microcapsules loaded with C5.18 chondrocytes alginate/chitosan prepared by a high-voltage electrostatic method. Under optimized conditions, microencapsulated cells presented uniform size distribution, good sphericity, and a smooth surface with different cell densities. The particle size distribution was determined at 150–280 μm, with an average particle diameter of 220 μm. The microencapsulated cells were cultured under static, shaking, and 3D micro-gravity conditions with or without bFGF (basic fibroblast growth factor) treatment. The quantified detection (cell proliferation detection and glycosaminoglycan (GAG)/type II collagen (Col-II)) content was respectively determined by cell counting kit-8 assay (CCK-8) and dimethylmethylene blue (DMB)/Col-II secretion determination) and qualitative detection (acridine orange/ethidium bromide, hematoxylin-eosin, alcian blue, safranin-O, and immunohistochemistry staining) of these microencapsulated cells were evaluated. Results showed that microencapsulated C5.18 cells under three-dimensional microgravity conditions promoted cells to form large cell aggregates within 20 days by using bFGF, which provided the possibility for cartilage tissue constructs in vitro. It could be found from the cell viability (cell proliferation) and synthesis (content of GAG and Col-II) results that microencapsulated cells had a better cell proliferation under 3D micro-gravity conditions using bFGF than under 2D conditions (including static and shaking conditions). We anticipate that these results will be a benefit for the design and construction of cartilage regeneration in future tissue engineering applications.

## 1. Introduction

The concepts of “cytomedicine”, tissue engineering, and artificial organs have been developed in recent years, in which the heterotypic contact, cell-to-cell signaling, and uniform nutrient diffusion are key points. Artificial cells, called microencapsulated cells, can provide a liquid environment for cell cultures and maintain cell phenotype. They have potential for the immobilization of cells and enzymes, artificial organ translation, tissue engineering applications, etc. [1] As presented in 1933, Bisceglie et al. sealed tumor cells in polymer membranes and transplanted them into the abdominal cavities of pigs. The results showed that the cells could survive long enough without being damaged by the immune system. Lim et al. [2] combined microencapsulation techniques with tissue-cell transplantation to prepare sodium alginate/poly-lysine/sodium alginate (APA) microcapsules with good biocompatibility as an immune isolation tool. As reported in research, the “artificial cells” embedded with porcine islet cells successfully regulated blood glucose levels and replaced pancreatic function in rats, called the “artificial pancreas”. This research has opened up a new method for cell transplantation therapy.

Current limitations of artificial cells cultured in vitro are mass-transfer, cell viability, and phenotype stability [3]. The microencapsulation of bioactive substances was first reported by Chang et al. in 1957. Microspheres prepared by hydrogel can provide a large surface area for cell growth and possess easy estimation for diffusion and mass-transfer behavior. Endres et al. [4] reported that 3-(methacryloxy) propyl trimethoxysilane (MPS) stem cells implanted in microcapsules could express chondrocytes which were induced by TGF-β3. Moreover, Chan et al. [4,5] prepared collagen microcapsules embedded in bone marrow mesenchymal stem cells and induced them to differentiate into chondrocytes, osteoblasts, and adipocytes in vitro. The results showed that cell encapsulation in hydrogels has been widely used in cytotherapy, regenerative medicine, 3D cell culture, and tissue engineering [6]. The immunological isolation of artificial cells can effectively avoid immune rejection induced by homologous cells, xenogeneic cells, and genetically engineered cells after transplantation in vivo, while allowing sufficient diffusion of oxygen, nutrients, secreted molecules, and metabolic waste for the cells to survive and perform their functions [7]. These results have broad application prospects in a variety of diseases and metabolic dysfunction treatments [8].

It is necessary to strengthen the good biocompatibility of microcapsules to successfully apply microencapsulation technology to the medical field [9]. Traditional construction has some disadvantages. For example, cells are sometimes difficult to grow into scaffolds and present uneven distribution. Different culture conditions can cause different viability in cells and further present different results in tissue engineering, which Liu has reported before [10]. Compared to 2D conditions, 3D cultures can protect chondrocyte models from species-specific acute drug toxicity in vitro and can also maintain cell phenotype. To date, interest has focused on fundamental research into microspheres cultured in rotary cell culture system (RCCS) conditions and their application. Also, a very effective growth factor, basic fibroblast growth factor (bFGF), that induces the proliferation of chondrocytes, as well as angiogenesis and wound healing by affecting smooth muscle cells, endothelial cells, fibroblasts, and epithelial cells, has been broadly applied in tissue engineering [11]. Studies have revealed that bFGF is beneficial for enhancing cell proliferation [12] and retaining chondrocytes’ phenotype [13].

The main components of the extracellular matrix are type II collagen, proteoglycans, glycosaminoglycans, hyaluronic acid, and some glycoproteins. The chondrocytes are terminally differentiated cells which are highly specific and their principal function is to maintain the stability of cartilage matrix components [14,15]. Articular cartilage may be damaged due to illness, trauma, or aging bodies, which determines the limitation of self-repairing capability. The demand for cartilage cells is rapidly increasing for cell-based research and medicine [16]. Herein, microencapsulated C5.18 cells were prepared by high-voltage electrostatic methods and the influences of different culture conditions with and without bFGF were investigated, which indicated a significant result in different culture conditions in vitro.

## 2. Materials and Methods

### 2.1. Materials

Sodium alginate was purchased from the National Pharmaceutical Group Chemical Reagent Co., Ltd. (Shanghai, China). Chitosan was obtained from Golden-Shell Biochemical Co., Ltd. (Zhejiang, China). Cysteine, hematoxylin and eosin (H&E) staining kit, papain, and safranin-O were kindly purchased from Solarbio Science and Technology Co., Ltd. (Beijing, China). Masson stain kit was obtained from Jiancheng Technology Co., Ltd. (Nanjing, China) C5.18 cells, acridine orange/ethidium bromide (AO/EB) and klcian blue staining Kit were provided by Syagen Biosciences Inc. (Guangzhou, China). Cell counting kit-8 (CCK-8) was purchased from Beyotime Biotechnology Co., Ltd. (Shanghai, China). 1,9-dimethylmethylene blue (DMB) was obtained from Sigma Co., Ltd. (St. Louis, MO, USA). Rat collagen type-II (Col-II) ELISA kit was purchased from Cusabio Biotech Co., Ltd. (Wuhan, China). Basic fibroblast growth factor (bFGF) was purchased from PeproTech Inc. (Rocky Hill, CT, USA).

### 2.2. Preparation of Blank microcapsules and Microencapsulated C5.18 Cells (Artificial Cells)

As shown in Figure 1, the blank microcapsules were prepared by using a WJN50 high-voltage electrostatic droplet generator (customized by Shanghai Ligong University) according to the work we reported elsewhere (parameters: 5.5 kV; 50 mm/h) [17]. Briefly, cells in the logarithmic phase were digested with trypsin and the required cell concentration was adjusted to 5 × 10^6^ cells/mL. The single cell suspension was firstly added into 2 mL of 1.5% alginate solution, which was further dropped into 1.5% CaCl_2_ solution by high-voltage electrostatic method (voltage: 5.5 kV; needle: 7^#^; distance of needle and CaCl_2_ solution: 2 cm; propulsion speed: 50 mm/h) to obtain alginate-calcium (Alg-Ca) beads loaded with C5.18 cells. In order to form alginate/chitosan (AC) film, C5.18 cell-loaded Alg-Ca beads were made up to 0.5% chitosan solution (pH 5.5) for a 10-min film reaction after being filtered by tea leakage and the beads with the chitosan (CS) film were transferred to sodium alginate solution to form an AC film [10]. Next, the liquefied core of the artificial cells was prepared with 1.6% sodium citrate solution for 2 min after being washed by saline. Those artificial C5.18 cells were finally diffused in complete medium in a culture flask with bFGF (100 ng/mL) under different conditions (static condition: 5% CO_2_, 37 °C, saturated humidity; shaking conditions: 5% CO_2_, 37 °C, saturated humidity, 60 r/min; rotary cell culture system: 5% CO_2_, 37 °C, saturated humidity, 60 r/min). Control groups were managed with no bFGF addition.

### 2.3. Characterization of Morphology

Blank gel beads and cell-laden microcapsules were dispersed in PBS (pH 7.4) buffer. The morphology and variation of particle size could be observed by an optical microscope (Olympus Corporation, Tokyo, Japan). The particle size was determined by reading the size of 500 particles. The surface morphology of the gel beads was observed by a scanning electron microscope (Hitachi, Tokyo, Japan) after a lyophilized management.

### 2.4. Cell Proliferation Assay

CCK-8 assay was used to determine the proliferation activity of microencapsulated C5.18 cells. The cell-laden gels were seeded in 96-well plates with 100 μL completed medium per well, then cultured in a 5% CO_2_ incubator at 37 °C with saturated humidity. Each group had three parallel tests. The optical density at 450 nm (OD_450_) value was directly in proportional to the number of C5.18 cells detected at vested culture time (1, 3, 5, 7, 10, 15, 20, and 30 days). 20 μL of 5 mg/mL CCK-8 was added into each well. After 4 h culturing, the 96-well plates were vibrated for 1 min using a microplate reader and the absorbance was measured at 450 nm using an enzyme-linked immunosorbent detector (Thermo, Waltham, MA, USA). The mean value was utilized to characterize the proliferation activity [18].

### 2.5. AO/EB (Acridine Orange/Ethidium Bromide) Staining

200 μL of cell-laden droplet suspension was incubated with 10 μL of AO/EB solution for 30 s. 10 μL of suspension was placed onto a microscopic slide covering by a glass coverslip and then the sample was observed using a fluorescence microscope (Nikon TI-S, Japan) using a fluorescein filter and a 60× objective [19].

### 2.6. Histological Staining

#### 2.6.1. Hematoxylin-Eosin Staining of Cell Section

Microencapsulated cells sections were fixed in 4% paraformaldehyde for 30 min. Samples were washed by PBS 3 times and then placed onto a microscopic slide covered by a glass coverslip. Hematoxylin solution was dripped onto the slides for 12 min. The slides were rinsed in H_2_O for 5 min and then were stained with 1% eosin Y solution for 5 min. The sections were dehydrated with two changes of 70% ethanol, two changes of 80% ethanol, and two changes of 100% ethanol for 30 sec each, then the ethanol was extracted with two changes of xylen for 5 min each. These slides were observed using confocal laser scanning microscopy (CLSM) (Leica company, Wetzlar, Germany) [20].

#### 2.6.2. Alcian Blue Stain

Microscope slides were incubated in 6-well plates with 400 μL laden-cells microcapsules suspension per well. After fixed in 4% paraformaldehyde for 30 min, the slides were stained with 1% alcian blue for 30 min. Later, the slides were respectively rinsed with PBS and absolute ethanol for 5 min and finally were observed using CLSM.

#### 2.6.3. Safranin-O

The hydrate slides were stained with safranin-O solution for 15 min and then washed in running tap water for 5 min. The slides were dehydrated with 95%, and100% alcohol for 20 sec each. Next, the slides were rinsed in xylen for 10 min. Samples were eventually observed under CLSM after being mounted in mounting medium.

### 2.7. Immunohistochemistry (IHC) Staining

Protocols for immunohistochemistry staining briefly were as follows: Each fixed section was placed in 0.5% triton X-100 solution for 20 min. After being fixed with 4% paraformaldehyde, the slides with cells were immersed in 0.5% triton X-100 for 20 min, then washed in PBS 3 times for 20 min each. After being incubated in 3% H_2_O_2_ for 15 min, the slides were washed with PBS 3 times for 2 min each and then were blocked in buffer containing 0.5% BSA and 2% FBS for 15 min. After that, the slides were placed in the diluted primary antibody (collagen II antibody) for 60 min at 37 °C, then were incubated with secondary antibody (horseradish peroxidase labeled goat anti-rabbit IgG) in a humidified chamber at room temperature. 20 min later, samples were rinsed with PBS 5 times for 2 min each and stained with 3,3′-diaminobenzidine (DAB) and hematoxylin for 10 min, respectively. Finally, the slides were washed in running water for later detection.

### 2.8. Glycosaminoglycan (GAG) Release [21]

Quantification of total glycosaminoglycans (GAG) in chondrocyte cultures is necessary for the complete assessment of the metabolic profile of the system and is commonly performed using the 1,9-dimethylmethylene blue (DMB) dye method [22,23]. The artificial cells obtained after determinate time were cultured with 300 μL papain in a water-bath at 60 °C overnight and then were centrifuged at 4000 rpm for 5 min to obtain supernatant samples. 100 μL of papain-digested samples were added into 2.5 mL DMB solution. Absorbance was determined at 525 nm. GAG content of the samples was determined using a C-6-S standard curve (0–100 μg/mL).

### 2.9. Type II Collagen Secretion

Artificial cells were washed by PBS. After being broken, suspension was centrifuged at 5000 rpm for 5 min to obtain subsidence-containing cells. Next, 1 mL of PBS was added and samples were stored at −20 °C overnight. After being centrifuged at 5000 rpm for 5 min, the obtained supernatant was detected at 450 nm using the type II collagen ELISA kit assay.

### 2.10. Statistical Analysis

Results were expressed as mean ± standard deviation. Statistical significance was measured by one-way ANOVA. * *p* < 0.05 and ** *p* < 0.01 were considered significant. Each measurement reported was based on duplicate analysis of at least three independent experiments.

## 3. Results and Discussion

### 3.1. Morphology of Microcapsules and Artificial Cells

As shown in Figure 2a, the blank alginate-chitosan microcapsules with a diameter of 150–280 μm were spherical and possessed a structure of liquid core which was suitable for cell cultivation [24]. The morphology of C5.18 cells encapsulated in microcapsules were observed in Figure 2b. The distribution of cells was uniform and viable. As shown in Figure 2c,d, the SEM observation of microcapsules presented a crude surface with multiple micro-holes which could decrease the resistance of mass transfer; the typical structure could supply a benign environment to culture cells in vitro [25].

### 3.2. Cell Viability

As shown in Figure 3a,b, microencapsulated cells stained in AO/EB were mostly green, which illustrated high cell viability. H&E staining related to the typical morphology of chondrocytes showed mostly purple, which indicated a superior status of cell proliferation. The results of AO/EB and H&E staining indicate that the microencapsulated operating process did not significantly affect cell viability, providing a promising capability of cartilage regeneration. The results illustrate that microcapsules could provide a 3D environment for cell growth, which restricts the entry of macromolecules and improves the absorption of nutrients to microcapsules.

### 3.3. Cell Proliferation Assay

Figure 4 illustrates that cell proliferation in 2D and 3D constructs with bFGF was greater than without bFGF. Studies have revealed that bFGF is beneficial for enhancing cell proliferation and retaining chondrocytes’ phenotype [26,27]. Under static conditions (Figure 4a), cells without bFGF attained their highest proliferation rate on day 7 (OD_450_ = 0.198), while they got to their peak (OD_450_ = 0.485) on day 10 with bFGF. Under shaking conditions (Figure 4b), the results showed their highest proliferation rate on day 10 when bFGF free (OD_450_ = 0.225), while they reached their peak on day 15 with bFGF (OD_450_ = 0.592), which presents a highly significant difference (*p* < 0.01). Cell proliferation was enhanced by bFGF and was significantly higher (*p* < 0.01) than without bFGF. The same trends were observed under RCCS conditions (Figure 4c). The cells showed similar proliferation rates in the 3D microgravity environment on day 15 without bFGF (OD_450_ = 0.225) and reached their highest proliferation on day 20 with bFGF (OD_450_ = 0.686), between which there existed a significant difference (*p* < 0.05). With the selective permeation of the microcapsule membrane, the substances with high molecular weight outside the microcapsule could not be diffused into the microcapsule and the nutrient components (bFGF) in the biological environment could freely enter the microcapsule, thus achieving good cell proliferation.

### 3.4. Concentration of GAG

Quantitative and qualitative results shown in Figure 5 indicate, under static conditions, the corresponding GAG of microencapsulated cells reached maximum (0.46 mg/mL) on day 7. The cells showed a clear advantage after bFGF was added and could reach the highest activity (0.77 mg/mL) on day 10 shown in Figure 5a_1_. Alcian blue and safranin-O staining results of chondrocytes on day 10 are presented in Figure 5a_2_, which shows positive staining. The results show that bFGF could improve the cells viability according to its mitosis-promoting capability [26]. As illustrated in Figure 5b_1_, under shaking conditions, the corresponding GAG values presented the maximum value (0.94 mg/mL) on day 10 with no bFGF and on day 15 with bFGF (1.33 mg/mL). The corresponding staining results in Figure 5b_2_ were considered the quantitative results of GAG. The results illustrated that the shaking environment had an advantage over the static environment. It was found from the content of GAG results in Figure 5c_1_, under 3D microgravity conditions, microencapsulated cells had maximum synthesis on day 20 after adding bFGF (1.86 mg/mL) and cells showed significant advantages compared with the control group that reached maximum synthesis on day 15 (1.54 mg/mL). The corresponding staining results (Figure 5c_2_) appeared positive staining and the cell aggregates were formed and grew to the maximum on day 20. These results illustrate that cells could obtain adequate nutrition to proliferate when incubated under the microgravity environment, which could protect cells existing in static and shaking conditions from obstacles and could increase the amount of Col-II content. C5.18 cell viability was enhanced by bFGF which could improve the activity of cell mitosis and the microgravity environment could also help chondrocytes keep their cartilage characteristics. Therefore, the number of cells was improved and the corresponding staining results were positive with large cell aggregates forming on day 20.

### 3.5. Type II Collagen Secretion

Static culture condition results were observed in Figure 6a. The quantitative results (Figure 6a_1_) of type II collagen synthesis of microencapsulated cells reached the maximum on day 5 without bFGF and the cells obtained the highest content of Col-II on day 10 after adding bFGF. Col-II immunohistochemical staining results of chondrocytes presented in Figure 6a_2_ were consistent with the quantitative results.

Some improvements were made according to the problems under static conditions. The prepared microencapsulated cells were cultured under shaking conditions. Cells in microcapsules obtained the maximum Col-II secretion and the corresponding positive staining on day 10 without bFGF. The maximum Col-II secretion of cells appeared on day 15, which was accelerated by bFGF, as shown in Figure 6b_1_. The positive staining indicated that the secretion amount of Col-II chondrocytes reached the maximum on day 15 (Figure 6b_2_), at the same time the cell-aggregate formation was observed.

Although cell aggregates were formed under shaking conditions and bFGF was added, they had not yet achieved the desired target of cartilage tissue construction in vitro, due to mass transfer still affecting the cartilage construction. The rotary cell culture system (RCCS) can provide 3D culture conditions to diffusive mass transfer, the feature necessary to elicit stability and functionality, requiring hetero-typic contact, cell-to-cell signaling, and uniform nutrient diffusion [28]. Therefore, microencapsulated cells were cultured under 3D micro-gravity conditions [9]. It was found from the synthesis content of Col-II results that, when there was no bFGF, microencapsulated cells had maximum synthesis on day 15 and after adding bFGF, cells showed significant advantages and reached the highest content of Col-II on day 20 (Figure 6c_1_). The corresponding staining results appeared positive within 20 days and the cell aggregates were formed and grew to the maximum on day 20. In Figure 6c_2_, the positive stain was in agreement with Col-II detection, which further illustrated that artificial cells had a higher synthesis content of Col-II within bFGF cultures in 3D conditions than that of 2D culture conditions.

## 4. Conclusions

Artificial cells were prepared using a high-voltage electrostatic method through microcapsule-loaded C5.18 chondrocytes using alginate and chitosan. Under optimized conditions, it obtained microencapsulated cells with uniform cell distribution, good sphericity, and a smooth surface by embedding a different cell density. The particle size distribution was 150–280 μm with an average particle diameter of 220 μm. AO/EB and H&E staining indicated high cell viability and the typical morphology of chondrocytes (time). The content of GAG and Col-II detection showed that the constructed C5.18 cells under 3D microgravity conditions could improve cells to form large cell aggregates with bFGF joining, which provides the possibility for cartilage tissue constructs in vitro.

Nutrients were placed under static environments and the presence of the chitosan membrane in microcapsules hindered communication between the cells and culture medium [29]. Furthermore, metabolic waste was present in the system at all times [30]. As a result, cells began to apoptose and die. Without the presence of bFGF, nutrition could get into microcapsules and constructs under dynamic conditions. Thus, cells under shaking and RCCS conditions survived longer than those under static conditions. However, the presence of chitosan still limited the contact between the cells and the culture medium. The presence of shear forces produced under dynamic conditions hindered cell growth, thus some cells started apoptosis. When bFGF was added, the values of the cells increased significantly, as bFGF could promote cell mitosis and therefore, cell proliferation. Cell mitosis and cellular uptake of nutrients could be promoted by bFGF and dynamic conditions, respectively, so the number of cells increased with increasing GAG and Col-II synthesis. Altogether, the cell viability and the corresponding amount of GAG and Col-II of chondrocytes were improved.

In summary, the results demonstrated that the microencapsulated C5.18 cells incubated in the medium containing bFGF under RCCS conditions could increase the number of cells with a chondrogenic phenotype, which further presents a method to obtain meaningful numbers of cells in artificial cells using the microencapsulation technique and provides a promising pathway to cartilage regeneration in vitro.

## Figures and Tables

**Figure 1 polymers-11-00471-f001:**
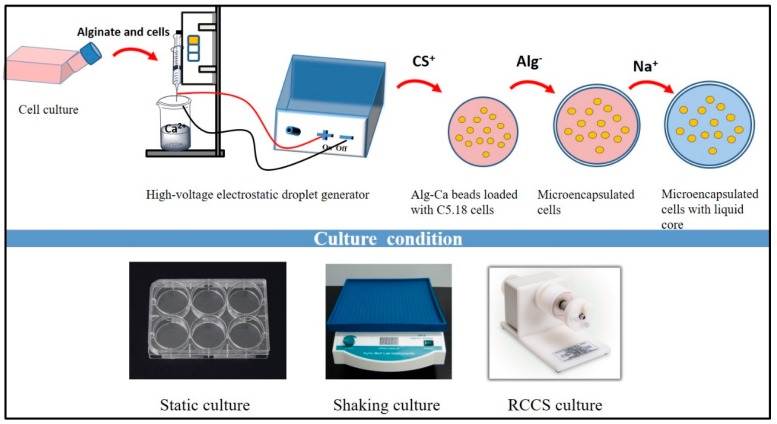
Schematic process of the preparation of artificial C5.18 cells.

**Figure 2 polymers-11-00471-f002:**
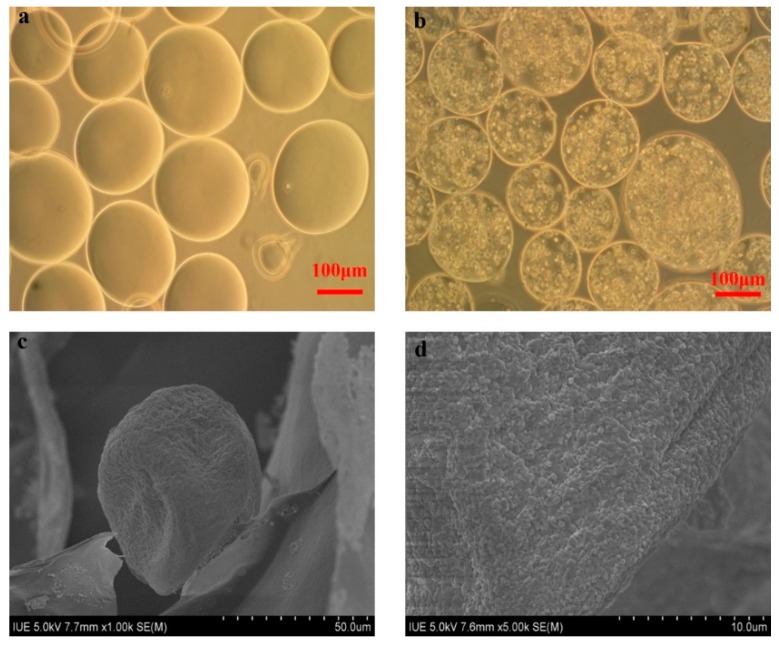
Morphology of blank microcapsules and artificial C5.18 cells. (**a**,**b**): Optical microscope of blank microcapsules and artificial C5.18 cells; (**c**,**d**): SEM images of microcapsules with 1000× and 5000× objective.

**Figure 3 polymers-11-00471-f003:**
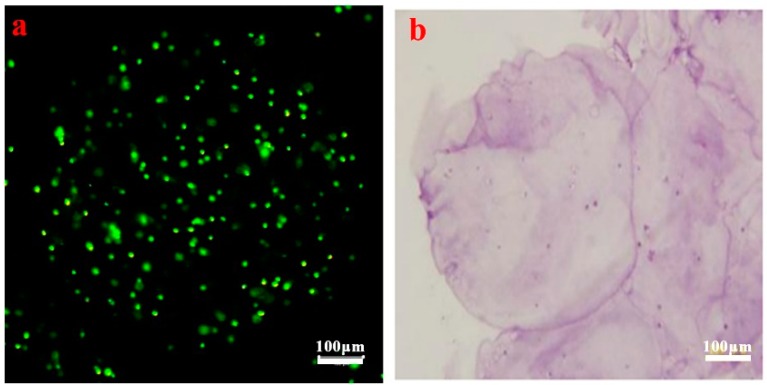
Confocal laser scanning microscopy (CLSM) image of artificial C5.18 cells managed by (**a**) acridine orange/ethidium bromide (AO/EB) staining and (**b**) hematoxylin and eosin (H&E) staining.

**Figure 4 polymers-11-00471-f004:**
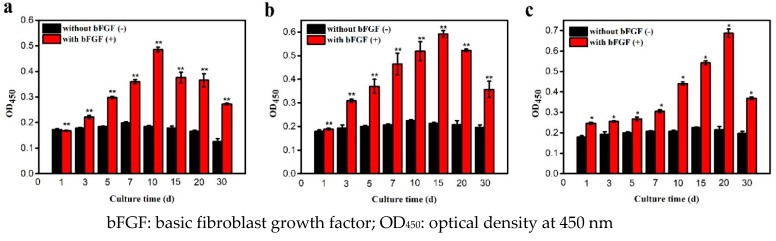
Proliferation of artificial cells under different culture conditions: (**a**), Static, (**b**) shaking, and (**c**) RCCS (** *p* < 0.01, * *p* < 0.05).

**Figure 5 polymers-11-00471-f005:**
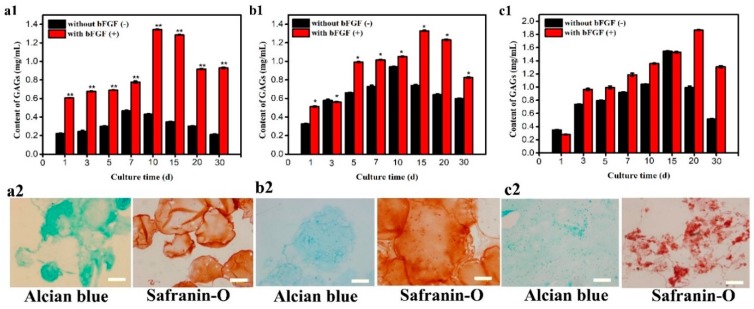
Glycosaminoglycan (GAG) content detection by 1,9-dimethylmethylene blue (DMB) under different culture conditions: (**a_1_**) Static, (**b_1_**) shaking, and (**c_1_**) RCCS (** *p* < 0.01, * *p* < 0.05); alcian blue staining and safranin-O staining of artificial C5.18 cells with bFGF under different culture conditions: (**a_2_**) Static, (**b_2_**) shaking, and (**c_2_**) RCCS. Bar is 100 μm.

**Figure 6 polymers-11-00471-f006:**
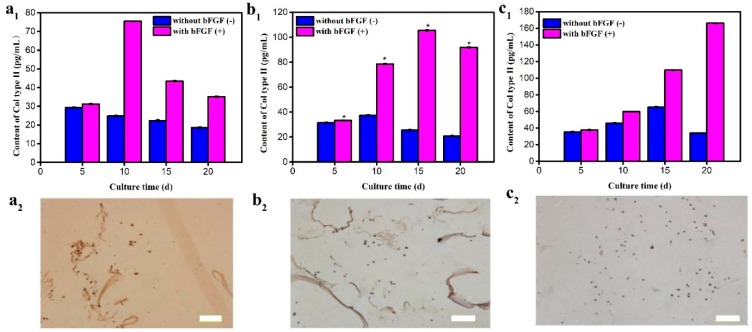
Type II collagen content detection under different culture conditions: (**a_1_**) Static, (**b_1_**) shaking, and (**c_1_**) RCCS (* *p* < 0.05); Immunohistochemistry staining of artificial C5.18 cells with bFGF under different culture conditions: (**a_2_**) Static, (**b_2_**) shaking, and (**c_2_**) RCCS. Bar is 100 μm.

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
