# Peer review of "The Influence of bFGF on the Fabrication of Microencapsulated Cartilage Cells under Different Shaking Modes"

_polymers, 2019, doi:10.3390/polym11030471_

Round 1
Reviewer 1 Report
The manuscript describes three preparation methods for encapsulating chondrocytes in microcapsules.
The major flaw of the manuscript is the lack of differentiation from already published work.
Acronyms and abbreviations should be minimized or spelled out int the abstract.
Introduction: the first two paragraphs are quite basic, it should describe latest advances in tissue engineering.
Not necessary to explain cartilage function at textbook level. references should also be pertinent (e.g., ref 1)
Lines 181-182: the statement about proliferation based on H&E staining is not accurate. Flow cytometry should be used quantitative assessment.
Figure 1 top, Figure 4a and c and Figure 5a and c: not clear what the difference is compared to date published in Ref 11.
Lines 241-243 confusing statement. In what way mass transfer affects cartilage production?
Discussion is largely missing. Not clear how these results compare to existing techniques in cartilage engineering and what the novelty is compared to previous studies, as the role of bFGF and culturing conditions have been already demonstrated.
Minor corrections:
Ref 11 and 13 are the same.
Author Response
Response to Reviewer 1 Comments
Reviewer 1#
Comments:
1. The manuscript describes three preparation methods for encapsulating chondrocytes in microcapsules. The major flaw of the manuscript is the lack of differentiation from already published work. Acronyms and abbreviations should be minimized or spelled out in the abstract. Introduction: the first two paragraphs are quite basic, it should describe latest advances in tissue engineering. Not necessary to explain cartilage function at textbook level. references should also be pertinent (e.g., ref 1).
Response: We authors sincerely appreciate the reviewer for the constructive comments. As presented, we have now revised the manuscript and supplied the details of Acronyms and abbreviations in abstract. And also, we have revised the first two paragraphs of the introduction with the latest advances in tissue engineering. Moreover, we delated the definition of cartilage function as well as changed the references according to the suggestions. Thanks.
“bFGF (basic fibroblast growth factor)
GAG (glycosaminoglycan)
Col-Ⅱ (type II collagen)
CCK-8 (cell counting kit-8)
DMB (dimethylmethylene blue)
AO/EB: acridine orange/ethidium bromide
H&E: hematoxylin-eosin”
2. Lines 181-182: the statement about proliferation based on H&E staining is not accurate. Flow cytometry should be used quantitative assessment. Figure 1 top, Figure 4a and c and Figure 5a and c: not clear what the difference is compared to date published in Ref 11. Lines 241-243 confusing statement. In what way mass transfer affects cartilage production?
Response: Many thanks for the comments. The detection of microencapsulated cells with H&E staining could indicate the viability of cells encapsulated in microcapsules which were consistent with the result of the observation of AO/EB. And also we have detected the cell proliferation via CCK-8 assay which further indicated good viability of cells cultured in microcapsules. As we published in reference 11, we fabricated the matrix-shaping with artificial cells, and in this paper we focused on the microencapsulated cells solely. As illustrated in figure 4a and 4c, under static conditions (figure 4a), cells without bFGF attained their highest proliferation rate on day 7 (OD450=0.198), while they got to their peak (OD450=0.485) on day 10 with bFGF. The cells showed similar proliferation rates in the three-dimensional microgravity environment on day 15 (OD450=0.225) cultured in bFGF-free medium, and reached their highest proliferation on day 20 (OD450=0.686) with bFGF, between which there existed significant difference. Comparing the results of GAG detection, as illustrated in figure 5a1, the contents of GAGs in microencapsulated cells reached the maximum value at 0.46 mg/mL in bFGF-free medium and at 0.77 mg/mL after adding bFGF under static culture condition. As presenting in figure 5c1, under the condition of RCCS, the content were 1.54 mg/mL and 1.86 mg/mL, respectively. These results illustrated that cells in microcapsules could obtain nutrition to proliferate and increase the amount of Col-II content when incubated under microgravity environment. When cells existed in static and shaking conditions, nutrition could be prevented from obstacle. More importantly, rotary cell culture system (RCCS) can provide 3D culture condition to diffusive mass transfer, the property which is necessary to elicit stability and functionality require hetero-typic contact, cell-to-cell signaling and uniform nutrient diffusion. Thus, mass transfer could affects cartilage production.
3. Discussion is largely missing. Not clear how these results compare to existing techniques in cartilage engineering and what the novelty is compared to previous studies, as the role of bFGF and culturing conditions have been already demonstrated. Minor corrections: Ref 11 and 13 are the same.
Response: We thank the reviewer for the thoughtful comment. As presented in ref 11, bFGF and 3D culturing conditions possessed a good achievement of cell culturing in 3D porous scaffolds. In this research, we focused on microencapsulated cells solely to confirm the cell culture conditions in vivo which has potential for artificial organic, tissue engineering as well as biomaterials application. We have corrected the same references: ref 10 and 13 in the revised manuscript, besides, we have supplemented more discussions in the revision. Thanks.

Reviewer 2 Report
The manuscript deals with the fabrication of artificial cells by encapsulation of C5.18 chondrocytes in alginate-chitosan microcapsules. Uniform cell distribution, good sphericity and smooth surface were detected by SEM analysis. Different culture conditions were investigated (with/without basic fibroblast growth factor –bFGF-; static and shaking conditions), pointing out that the chondrocyte proliferation was enhanced by adding bFGF, that improves the cell mitosis forming large cell aggregates, with bFGF joining.
Moreover, under shaking conditions an higher proliferation rate was observed.
The topic is interesting for tissue engineering specialists and it provides a promising way for cartilage regeneration in vitro. In the title, I suggest replacing the misleading word “function” with “fabrication”.
In my opinion, the paper is acceptable for publication, after minor revision, as follow:
- The acronym bFGF should be explained the first time it appears in the abstract (line 19).
- page 2, line 49: “….old. And….” : please check the sentences.
- page 5, lines 171-174: the sentence should be rewritten.
-page 6, line 206: please delete the bracket.
Author Response
Response to Reviewer 2 Comments
Reviewer 2#:
Comments:
The manuscript deals with the fabrication of artificial cells by encapsulation of C5.18 chondrocytes in alginate-chitosan microcapsules. Uniform cell distribution, good sphericity and smooth surface were detected by SEM analysis. Different culture conditions were investigated (with/without basic fibroblast growth factor –bFGF-; static and shaking conditions), pointing out that the chondrocyte proliferation was enhanced by adding bFGF, that improves the cell mitosis forming large cell aggregates, with bFGF joining. Moreover, under shaking conditions a higher proliferation rate was observed. The topic is interesting for tissue engineering specialists and it provides a promising way for cartilage regeneration in vitro. In the title, I suggest replacing the misleading word “function” with “fabrication”.
In my opinion, the paper is acceptable for publication, after minor revision, as follow:
- The acronym bFGF should be explained the first time it appears in the abstract (line 19).
- page 2, line 49: “….old. And….” : please check the sentences.
- page 5, lines 171-174: the sentence should be rewritten.
-page 6, line 206: please delete the bracket.
Response: We authors appreciate the reviewer for the suggestion and we apologize for the inconvenience caused during the peer review. As suggested, we have revised the title from “the influence of bFGF on the function of artificial cartilage cells under different shaking modes” to “the influence of bFGF on the fabrication of microencapsulated cartilage cells under different shaking modes”. Furthermore, we filled the full name of bFGF in the abstract. Moreover, we rewritten some confusing sentences following the suggestion presented in section 1 “Introduction”, section 3.1 “Morphology of Microcapsules and Artificial Cells” as well as deleted the bracket of the sentence in section 3.4 “Concentration of GAG”. Thanks.

Reviewer 3 Report
Overall, this is a fairly straightforward study that is rationally presented. In addition to English language editing, the following changes are recommended prior to publication:
1. The term ‘artificial cells’ seems to be mis-used in the manuscript. Please consider a different terminology or provide a better definition for artificial cells and how your work relates to the definition, as the cells in your experiments are normal/real cells.
2. Section 2.7: Specify the primary and secondary antibodies used for IHC staining.
3. Figure 4: It is difficult to distinguish between the pattern fillings for +/- bFGF. Please change the fillings to have more contrast and increase the font size on these charts.
4. Figures 5 and 6: Change the bar fillings to improve contrast and increase font size.
Author Response
Response to Reviewer 3 Comments
Reviewer 3#:
Comments:
Overall, this is a fairly straightforward study that is rationally presented. In addition to English language editing, the following changes are recommended prior to publication:
1. The term ‘artificial cells’ seems to be mis-used in the manuscript. Please consider a different terminology or provide a better definition for artificial cells and how your work relates to the definition, as the cells in your experiments are normal/real cells.
Response: Many thanks for the thoughtful suggestion. Artificial cells, called microencapsulated cells, can provide a liquid environment for cells culture as well as maintain cell phenotype. As suggested, we have changed the “artificial cells” into “microencapsulated cells” in title which were fabricated by microcapsules loaded with C5.18 chondrocytes alginate/chitosan through high-voltage electrostatic method.
2. Section 2.7: Specify the primary and secondary antibodies used for IHC staining.
Response: We thank the reviewer for the comments. We supplemented the primary and secondary antibody used for IHC staining in section 2.7, which was collagen II antibody and horseradish peroxidase labeled goat anti-rabbit IgG, respectively. Thanks.
3. Figure 4: It is difficult to distinguish between the pattern fillings for +/- bFGF. Please change the fillings to have more contrast and increase the font size on these charts.
Response: Many thanks for the reviewer’s comments and we apologize the inconvenience caused during the peer review. We have now revised Figure 4 in the revision manuscript.
4. Figures 5 and 6: Change the bar fillings to improve contrast and increase font size.
Response: Thank you very much for the valuable comment. As presented in figure 5 and 6, we have revised figures according to the suggestion. Thanks.

Round 2
Reviewer 1 Report
the authors addressed the concerns